# FMDV VP3 Induces IL-10 Expression in Porcine Macrophages via PI3K Interaction and PI3K/AKT-mTOR Pathway Activation

**DOI:** 10.3390/v18010066

**Published:** 2026-01-01

**Authors:** Yuling Li, Zijing Guo, Yan Zhang, Li Luo, Chunsai He, Qiqi Xia, Jingyuan Zhang, Zhidong Zhang, Yanmin Li

**Affiliations:** Key Laboratory of Animal Medicine of Sichuan Education Department, College of Animal and Veterinary Sciences, Southwest Minzu University, Chengdu 610041, China; liyuling1401@163.com (Y.L.); zijingguo7@163.com (Z.G.); yanz0227@163.com (Y.Z.); 18783416104@163.com (L.L.); hcs175017@163.com (C.H.); xia77_rain@163.com (Q.X.); zhangjingyuan1007@163.com (J.Z.); liyanmin@swun.edu.cn (Y.L.)

**Keywords:** IL-10, FMDV VP3, PI3K/AKT-mTOR pathway

## Abstract

Foot-and-mouth disease virus (FMDV) infection elicits sustained, high-level interleukin-10 (IL-10) secretion in cattle and pigs, which correlates with lymphopenia and immunosuppression. We previously showed that macrophages are the principal source of IL-10 during FMDV infection in mice, but the viral trigger and host pathways remained unknown. In the present study, we examined whether the FMDV structural protein VP3 regulates IL-10 expression. To this end, a eukaryotic VP3 expression vector was transfected into porcine alveolar macrophages (3D4/21 cells), and IL-10 expression together with related signaling pathways was interrogated by qRT-PCR, ELISA, Western blot, co-immunoprecipitation (Co-IP), confocal microscopy, and luciferase reporter assays. The results showed that VP3 significantly increased IL-10 mRNA and protein levels (*p* < 0.001) in a time-dependent manner. Mechanistically, VP3 promoted phosphorylation of PI3K, AKT, and mTOR; this effect was abolished by the PI3K inhibitor LY294002, which also abrogated VP3-induced IL-10 secretion (*p* < 0.05). Furthermore, VP3 upregulated mRNA expression of STAT3, ATF1, and CREB (*p* < 0.05) and enhanced IL-10 promoter activity. The STAT3 inhibitor Stattic reduced IL-10 secretion by 22% (*p* < 0.05). Co-IP and confocal microscopy confirmed direct binding of VP3 to PI3K in the cytoplasm. In conclusion, FMDV VP3 induces IL-10 overexpression by directly activating the PI3K/AKT-mTOR signaling pathway, thereby elucidating a key mechanism of FMDV-induced immunosuppression.

## 1. Introduction

Foot-and-mouth disease virus (FMDV), a member of the genus *Aphthovirus* within the family *Picornaviridae*, causes a highly contagious vesicular disease in cloven-hoofed animals, including cattle, sheep, and pigs, leading to substantial economic losses in the livestock industry [1]. The FMDV genome comprises a single-stranded, positive-sense RNA molecule of approximately 8.5 kb [2], which is translated into a large polyprotein precursor. This precursor is subsequently processed by viral proteases into four structural proteins (VP1–VP4) that form the viral capsid, as well as eight non-structural proteins (Lpro, 2A, 2B, 2C, 3A, 3B, 3C, 3D) required for viral replication [3]. Among these viral proteins, VP3 plays essential roles in viral infection and host immune modulation [4], including the sequestration of mitochondrial antiviral signaling protein (MAVS) to suppress type I interferon induction [5] and the degradation of JAK1 to inhibit IFN-γ signaling [6].

Interleukin-10 (IL-10) is a potent immunoregulatory cytokine produced by activated immune cells [7,8] with critical roles in the immune response to viral infection [9,10]. By restraining T-cell expansion and suppressing the secretion of pro-inflammatory cytokines such as IL-1β and TNF-α, IL-10 negatively regulates the antiviral response [11,12,13]. FMDV exemplifies this strategy. Previous studies have shown that FMDV infection induces sustained, high-level IL-10 secretion in cattle and pigs, with a peak observed at days 3–4 post-infection [14]. This elevated IL-10 correlates with lymphopenia (depletion of peripheral blood lymphocytes) and a higher probability of establishing persistent infection in cattle [15]. Using a murine model, we have further demonstrated that the elevated serum IL-10 levels observed in FMDV-infected mice correlate with lymphopenia. IL-10 knockout (IL-10^-/-^) mice or in vivo blocking of IL-10/IL-10R signaling alleviates FMDV-induced lymphopenia, highlighting IL-10 as a central regulator of this pathological process. Furthermore, we identified macrophages as the dominant source of IL-10 during acute infection, and FMDV infection of mouse bone marrow-derived macrophages (BMDMs) in vitro triggers a robust IL-10 response [16]. However, the viral proteins responsible for IL-10 induction in macrophages and the underlying host signaling pathways remain undefined.

In the present study, we demonstrated that FMDV VP3 triggered robust IL-10 mRNA expression and protein secretion in porcine alveolar macrophages (3D4/21 cells). FMDV VP3 physically interacts with PI3K, activating the PI3K/AKT–mTOR signaling cascade and downstream phosphorylation of the transcription factors STAT3, ATF1 and CREB. These data uncover a direct link between a viral capsid protein and the induction of an immunosuppressive cytokine, and provide insight into the immunopathogenesis of FMDV-related lymphopenia.

## 2. Materials and Methods

### 2.1. Cell and Viral Plasmids

The porcine alveolar macrophage cell line (3D4/21) was routinely maintained in our laboratory. The cell line was cultured in RPMI 1640 medium (Gibco, Thermo Fisher Scientific, Waltham, MA, USA; Catalog No. 11875093) supplemented with 10% FBS (Bio-Channel, Nanjing, China; Catalog No. BC-SE-FBS07). The pcDNA3.1-3×FLAG-VP3 recombinant plasmid (GenBank accession number: JN998085) and the empty vector pcDNA3.1-3×FLAG (control) were purchased from Wuhan GeneCreate Biological Engineering Co., Ltd., Wuhan, China. The pGL-IL-10 and pRL-TK plasmids were purchased from Tsingke Biological Engineering Co., Ltd., Beijing, China. Plasmid sequencing (Sangon Biotech, Shanghai, China) confirmed the sequence integrity.

### 2.2. Plasmid Transfection

Exponentially growing 3D4/21 cells were plated into 6-well plates and incubated at 37 °C in an atmosphere of 5% CO_2_ until they reached approximately 70–80% confluence. In accordance with the manufacturer’s protocol for the Lipo6000™ transfection reagent (Beyotime, Shanghai, China; Catalog No. C0526), cells were transiently transfected with either the pcDNA3.1-3×FLAG-VP3 expression construct or the corresponding empty pcDNA3.1-3×FLAG vector. After transfection, cultures were maintained, and both cells and their culture supernatants were harvested at 12, 24, and 36 h for subsequent assessment of gene expression and protein levels.

### 2.3. RT-qPCR

Cells were plated in appropriate culture dishes and incubated at 37 °C in a humidified atmosphere containing 5% CO_2_ until they reached the desired confluence. Total RNA was isolated using RNAiso Plus (Takara Biomedical Technology, Beijing, China; Catalog No. 9108) according to the manufacturer’s instructions, and RNA concentration and purity were assessed by spectrophotometry before downstream analyses. The purified RNA was subsequently reverse-transcribed into cDNA using MetaScript qPCR RT SuperMix (Exongen, Xiamen, China; Catalog No. A504-01) to eliminate residual genomic DNA. Quantitative real-time PCR was performed using UltraStart Universal SYBR qPCR MasterMix (Exongen, Xiamen, China; Catalog No. A411-02) on a real-time PCR system following the recommended cycling parameters. All experiments were conducted in triplicate with independent biological samples, and the resulting data are presented as the mean ± standard deviation.

### 2.4. Signaling Pathway Inhibition

To investigate the roles of specific signaling pathways, 3D4/21 cells were pretreated with pathway-specific inhibitors. For PI3K inhibition, cells were treated with 10 µM LY294002 (MedChemExpress, Monmouth Junction, NJ, USA; Catalog No. HY-10108) or left untreated for 1 h. For STAT3 inhibition, a separate set of cells was treated with 20 µM Stattic (MedChemExpress, Monmouth Junction, NJ, USA; Catalog No. HY-13818) or left untreated for 24 h. Following these pretreatment regimens, all groups were transfected with the foot-and-mouth disease virus VP3 (FMDV VP3) recombinant plasmid. LY294002 is a broad-spectrum class I PI3K inhibitor, and Stattic is a selective STAT3 inhibitor.

### 2.5. Western Blot Analysis

Cells were lysed in RIPA buffer (Solarbio, Beijing, China; Catalog No. R0010) containing PMSF (Solarbio, Beijing, China; Catalog No. P0100) and Phosphatase Inhibitor II Cocktail (Life-iLab, Shanghai, China; Catalog No. AP03L025). After centrifugation, supernatants were denatured with loading buffer (Solarbio, Beijing, China; Catalog No. S1010) and proteins were separated by SDS-PAGE. Following transfer to PVDF membranes, blocking was performed with 5% skim milk. Membranes were incubated overnight at 4°C with primary antibodies: anti-PI3K p85 (HuaAn Biotechnology, Hangzhou, China; Catalog No. ET-1602-4), anti-phospho-PI3K p85 (Tyr458) (HuaAn Biotechnology, Hangzhou, China; Catalog No. ET-1603-4), anti-mTOR (Proteintech, Wuhan, China; Catalog No. 66888-1-Ig), anti-phospho-mTOR (Ser2448) (Proteintech, Wuhan, China; Catalog No. 67778-1-Ig), anti-Akt (Cell Signaling Technology, Danvers, MA, USA; Catalog No. 4691), anti-phospho-Akt (Ser473) (Cell Signaling Technology, Danvers, MA, USA; Catalog No. 4060), and anti-β-Actin (ABclonal, Wuhan, China; Catalog No. AC026). After TBST washes, membranes were incubated with HRP-conjugated secondary antibodies and signals were developed using enhanced chemiluminescence substrate (Affinity Biosciences, Cincinnati, OH, USA; Catalog No. AF-2805) with a BLT GelView6000Plus imaging system.

### 2.6. ELISA for Detection

3D4/21 cells were seeded in 6-well plates and cultured overnight. After washing with PBS, the cells were transfected with FMDV VP3 and incubated at 37 °C with 5% CO_2_. Supernatants were collected at 12, 24, and 36 h post-transfection, with the maintenance medium replaced after each collection. IL-10 concentrations in the supernatants were quantified using a commercial ELISA kit (COIBO BIO, Shanghai, China; Catalog No. CB10033-Pg). The optical density at 450 nm (OD_450_) was measured using an Epoch2 microplate reader. A standard curve was generated by measuring a dilution series of IL-10 standards, followed by linear regression analysis after blank subtraction. Statistical analysis was then performed to evaluate differences between experimental groups.

### 2.7. Co-IP

3D4/21 cells were transfected with FMDV VP3 for 24 h, and co-immunoprecipitation (Co-IP) was performed using the Magnetic IP/Co-IP Kit (Vazyme, Nanjing, China; Catalog No. FD8110). Cells were lysed for 20 min at 4 °C, and the lysates were centrifuged at 13,000× *g* for 10 min. The resulting supernatant was collected, and a portion was saved as the input control. The remaining lysate was incubated with 5–10 µg of specific primary antibody overnight at 4 °C. Then, 25 µL of Pierce Protein A/G Magnetic Beads were added to the mixture and incubated at room temperature for 1 h. The beads were collected using a magnetic rack, washed three times, and eluted with elution buffer. Finally, protein interactions were detected by Western blotting.

### 2.8. Immunofluorescence and Confocal Microscopy

3D4/21 cells were cultured on poly-L-lysine-coated confocal dishes. Once the cells had attached, they were transfected with the eukaryotic expression plasmid encoding FMDV VP3 and maintained for a further 24 h. Subsequently, the cells were fixed in 4% paraformaldehyde for 20 min, treated with 0.2% Triton X-100 for 15 min to permeabilize the membranes, and incubated in 5% BSA for 1 h to block nonspecific binding. The samples were then incubated with the corresponding primary antibody at 4 °C overnight, followed by a 1 h incubation at room temperature in the dark with a fluorophore-labeled secondary antibody. Nuclear DNA was counterstained with DAPI (Abcam, Cambridge, UK; Catalog No. ab228549) for 5 min, and images were acquired using a confocal laser scanning microscope.

### 2.9. Dual Fluorescein Reporter Gene Detection of IL-10 Promoter Activity

First, cells were co-transfected with pGL-IL-10 and pRL-TK plasmids. After 24 h, cells were transfected with FMDV-VP3 plasmid. After another 24-h incubation, the cells were collected, and IL-10 promoter activity was measured using a Dual-Luciferase Assay Kit (Beyotime, Shanghai, China; Catalog No. RG027).

### 2.10. Data Analysis

All quantitative data are reported as the mean ± standard deviation (SD) from at least three independent experiments. Statistical procedures were carried out using GraphPad Prism (version 10). For comparisons between two groups, unpaired Student’s *t*-tests were applied. When more than two groups were involved, data were analyzed by one-way analysis of variance (ANOVA) followed by Bonferroni’s multiple-comparison post hoc test. Values with *p* < 0.05 were regarded as statistically significant, whereas *p* < 0.01 was considered to indicate a highly significant difference.

## 3. Results

### 3.1. FMDV VP3 Induces High Level of IL-10 Expression in Porcine Alveolar Macrophages

To investigate whether FMDV VP3 induces IL-10 expression, porcine alveolar macrophages (3D4/21 cells) were transfected with a eukaryotic VP3-expression plasmid or an empty vector. As shown in Figure 1b, the IL-10 protein level remained at baseline in both groups at 12 h post-transfection, with no significant difference observed (*p* > 0.05). However, by 24 h and 36 h, cell supernatants from VP3-transfected cells exhibited a marked increase in IL-10 secretion, reaching concentrations of 69.4 pg/mL and 68.1 pg/mL, respectively. These levels were significantly higher than those in the vector controls (*p* < 0.01). In parallel, transcript-level analysis by qPCR confirmed this time-dependent induction and revealed an earlier onset of upregulation. As shown in Figure 1c, no significant upregulation of IL-10 mRNA was observed in VP3-expressing cells at 12 hpt compared to controls (*p* > 0.05), and this upregulation was sustained at 24 hpt (*p* < 0.0001) and 36 hpt (*p* < 0.0001). These data demonstrate that FMDV VP3 induces sustained, high-level IL-10 expression in 3D4/21 macrophages.

### 3.2. FMDV VP3 Activates the PI3K/AKT-mTOR Pathway in 3D4/21 Macrophages

The PI3K/AKT-mTOR signaling axis is a well-established regulator of IL-10 secretion in macrophages. Mechanistically, activated PI3K phosphorylates AKT, which in turn phosphorylates mTOR; this sequential activation promotes the nuclear translocation and binding of transcription factors (STAT3, ATF1, and CREB) to the IL-10 promoter, thereby initiating IL-10 transcription. To determine whether FMDV VP3 induces IL-10 expression through this pathway, we transfected 3D4/21 cells with a eukaryotic VP3-expression plasmid or an empty vector. Western blotting revealed a significant increase in the phosphorylation of key pathway components—PI3K, AKT, and mTOR—in VP3-expressing cells compared to vector controls at 24 hpt (Figure 2). Specifically, the levels of p-PI3K, p-AKT, and p-mTOR were robustly upregulated. These findings confirm that FMDV VP3 activates the PI3K/AKT-mTOR signaling cascade in 3D4/21 cells.

### 3.3. Inhibition of PI3K Phosphorylation Attenuates VP3-Induced IL-10 Expression in 3D4/21 Macrophages

To determine whether PI3K is required for VP3-induced IL-10 expression, 3D4/21 cells were pre-treated with the specific PI3K inhibitor LY294002 prior to transfection with the VP3-expression plasmid. Cells were harvested at 24 hpt for analysis of pathway activation and IL-10 secretion. Western blotting confirmed that LY294002 effectively abrogated VP3-induced phosphorylation of PI3K (p-PI3K). The inhibition of this upstream kinase consequently suppressed the phosphorylation of downstream effectors AKT and mTOR (p-AKT and p-mTOR), whereas total protein levels remained unaffected across groups (Figure 3a). This result indicates that PI3K inhibition blocks VP3-induced activation of the entire PI3K/AKT-mTOR cascade. Consistently, ELISA analysis of cell supernatants revealed that LY294002 pre-treatment significantly attenuated VP3-induced IL-10 secretion. IL-10 levels in inhibitor-treated cells were reduced by 26% compared to VP3-only controls (*p* < 0.05, Figure 3b). Notably, IL-10 secretion in LY294002+VP3 cells was comparable to that in vector-transfected controls, demonstrating that PI3K activity is indispensable for VP3-driven IL-10 production.

### 3.4. FMDV VP3 Interacts with PI3K

To investigate whether VP3 stimulates the PI3K/AKT-mTOR pathway through physical interaction with PI3K, 3D4/21 macrophages were transfected with Flag-tagged FMDV VP3. At 24 h post-transfection, cell lysates were subjected to co-immunoprecipitation (Co-IP). The results showed that endogenous PI3K was specifically co-precipitated with Flag-VP3, but not with the Flag-vector control, Conversely, Flag-VP3 was reciprocally co-precipitated with PI3K, confirming the specificity of the interaction (Figure 4a). Furthermore, confocal imaging revealed punctate cytoplasmic co-localization of VP3 (red) and PI3K (green) in the cells (Figure 4b). These data demonstrate that FMDV VP3 physically interacts with PI3K in 3D4/21 macrophages.

### 3.5. FMDV VP3 Regulates Transcription Factors STAT3, ATF1, and CREB to Drive IL-10 Promoter Activity

To elucidate the mechanism by which FMDV VP3 links the PI3K/AKT-mTOR signaling pathway to IL-10 transcription, we investigated its regulation of downstream transcription factors (STAT3, ATF1, CREB, and c-FOS) known to associate with IL-10 expression. 3D4/21 macrophages were transfected with FMDV VP3, and after 24 h, qPCR was used to quantify the mRNA levels of these four transcription factors, The detailed primers for qPCR were in Table 1. Compared to the vector control group, the mRNA expression of STAT3, ATF1, and CREB was significantly increased (all *p* < 0.01; Figure 5a). Among these, STAT3 induction was the most robust (*p* < 0.001), while c-FOS expression remained unchanged. To functionally validate the role of STAT3, we first confirmed elevated total and phosphorylated STAT3 levels in FMDV VP3-transfected cells (Figure 5b). We then inhibited STAT3 pharmacologically using Stattic, a specific inhibitor of STAT3 phosphorylation. Western blot analysis showed that Stattic abolished VP3-induced STAT3 phosphorylation (Figure 5c) and led to a 22% reduction in secreted IL-10 levels (*p* < 0.05; Figure 5d), thereby directly linking STAT3 activation to IL-10 production. Furthermore, a dual-luciferase reporter assay demonstrated that IL-10 promoter activity was significantly enhanced in the FMDV VP3 transfection group compared to the empty vector control (*p* < 0.01; Figure 5e). These results collectively confirm that VP3-driven upregulation of STAT3, ATF1, and CREB enhances IL-10 transcriptional activity.

## 4. Discussion

FMDV infection induces sustained, high-level IL-10 secretion in cattle, pigs, and mice, which correlates with lymphopenia and immunosuppression [17]; however, the underlying viral determinants and signaling pathways have remained undefined. IL-10, a classical anti-inflammatory cytokine, dampens host antiviral responses by inhibiting pro-inflammatory cytokine production (IL-1β, TNF-α) and impairing adaptive immunity (T-cell activation, B-cell antibody secretion) [18]. Our findings identify VP3 as a critical inducer of IL-10 in macrophages, complementing previous reports that VP3 inhibits type I interferon signaling via MAVS sequestration and JAK1 degradation [19,20]. Thereby, this dual role of VP3—suppressing antiviral immunity while promoting an immunosuppressive cytokine—underscores its central role in FMDV pathogenesis [21]. While FMDV non-structural proteins such as the 3C protease have been implicated in cytokine dysregulation [22], our study is the first to link a structural protein (VP3) to IL-10 regulation, thereby broadening the known repertoire of FMDV immune evasion strategies.

The PI3K/AKT-mTOR pathway is a conserved regulator of immune response, including IL-10 production [23]. The activation of this pathway leads to the binding of specific transcription factors to the IL-10 promoter to initiate transcription [24]. Our data extend this paradigm by demonstrating that VP3 directly interacts with PI3K to trigger pathway activation, a mechanism that distinguishes it from other viruses like PRRSV, which activate PI3K indirectly via receptor-mediated signaling. This direct interaction may enable VP3 to hijack host signaling more efficiently. Future studies employing mutational analyses (e.g., truncation mutants) are needed to map the specific VP3 residues required for PI3K binding. We further showed that VP3 upregulates the expression and activity of the transcription factors STAT3, ATF1, and CREB, all of which are known to bind the IL-10 promoter and drive its transcription. Among them, STAT3 acts as the core driver, while CREB and ATF1 act as co-activators to enhance transcriptional efficiency. The finding that inhibition of STAT3 with Stattic significantly reduced VP3-induced IL-10 secretion underscores the non-redundant, critical role of STAT3 in this process. This regulatory model is consistent with reports that PI3K/AKT-mTOR signaling can coordinately activate multiple transcription factors to fine-tune cytokine expression [25].

Elevated IL-10 during FMDV infection is associated with lymphopenia and persistent infection. By inducing IL-10, VP3 may impair T-cell proliferation and macrophage activation, thereby creating an environment permissive for viral replication and persistence. This model is supported by our earlier findings that IL-10 knockout mice exhibit reduced FMDV-induced lymphopenia, and that high early-stage IL-10 levels in cattle predict persistent infection. Future in vivo validation using susceptible animal models will be essential to confirm the physiological relevance of these findings.

## 5. Conclusions

FMDV VP3 induces IL-10 overexpression in macrophages (3D4/21 cells) by directly activating the PI3K/AKT-mTOR signaling pathway. These findings reveal a novel mechanism underlying FMDV-induced IL-10 expression and provide insight into the immunopathogenesis of FMDV-related lymphopenia and immunosuppression.

## Figures and Tables

**Figure 1 viruses-18-00066-f001:**
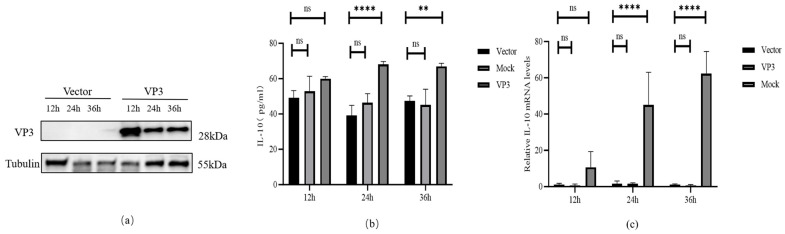
FMDV VP3 transfection induces IL-10 expression in 3D4/21 cells: (**a**) VP3 expression confirmed by Western blot; (**b**) IL-10 secretion in supernatants measured by ELISA at 12, 24, and 36 h post-transfection; (**c**) IL-10 mRNA levels quantified by qPCR at 12, 24, and 36 h post-transfection. ns (not significant) *p* > 0.05; ** *p* ≤ 0.01; **** *p* ≤ 0.0001.

**Figure 2 viruses-18-00066-f002:**
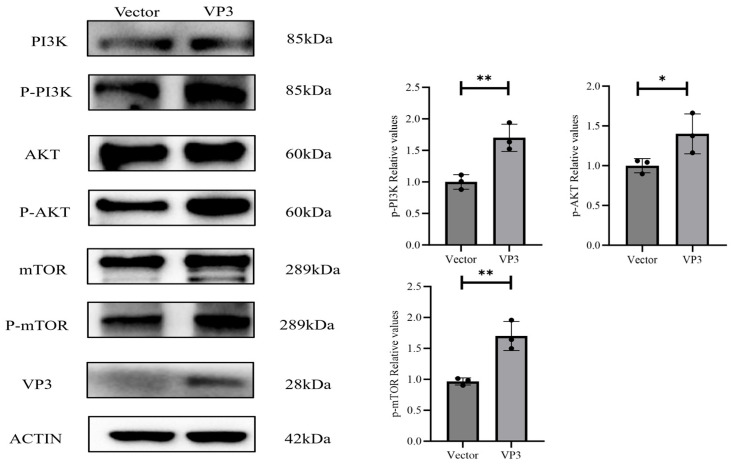
FMDV VP3 activates the PI3K/AKT-mTOR pathway in 3D4/21 cells. Western blot analysis of PI3K, AKT, and mTOR and their phosphorylated forms (p-PI3K, p-AKT, p-mTOR) in cells transfected with FMDV VP3 or empty vector for 24 h. VP3 expression enhanced the phosphorylation of all three key components in the PI3K/AKT-mTOR pathway. * *p* ≤ 0.05; ** *p* ≤ 0.01.

**Figure 3 viruses-18-00066-f003:**
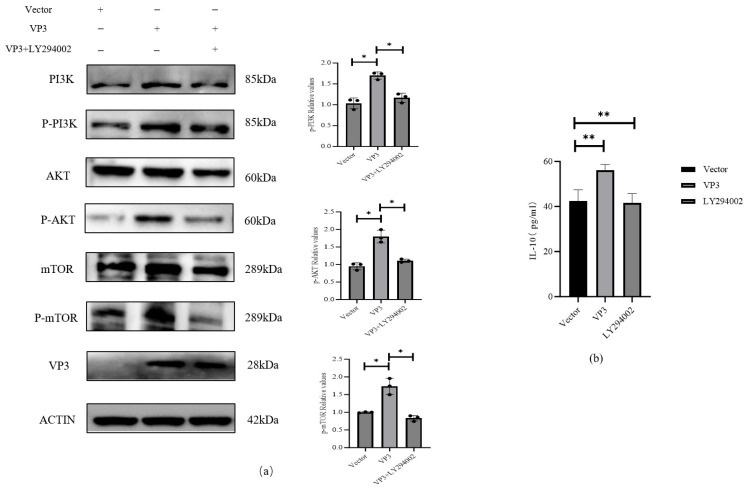
PI3K inhibition reduces VP3-induced IL-10 expression in 3D4/21 cells: (**a**) LY294002 suppresses VP3-induced PI3K/AKT/mTOR phosphorylation (Western blot, 24 h); (**b**) IL-10 secretion decreases with LY294002 treatment (ELISA; mean ± SD, *n* = 3); * *p* < 0.05, ** *p* < 0.01.

**Figure 4 viruses-18-00066-f004:**
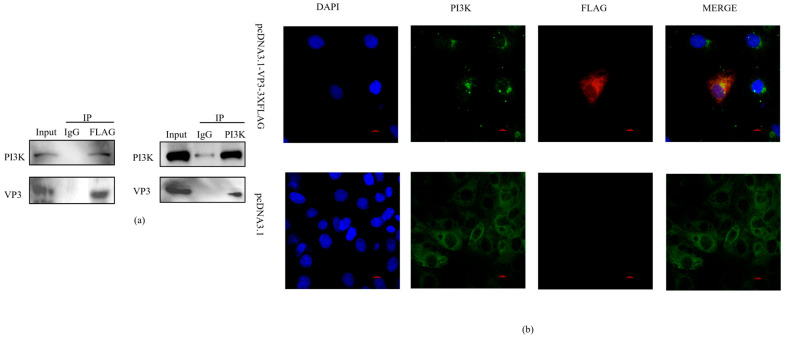
FMDV VP3 interacts with PI3K in 3D4/21 cells: (**a**) Co-immunoprecipitation of Flag-VP3 with endogenous PI3K. (**b**) Confocal microscopy showing cytoplasmic co-localization of VP3 and PI3K at 24 h post-transfection (DAPI-stained nuclei). Scale bar: 10 μm.

**Figure 5 viruses-18-00066-f005:**
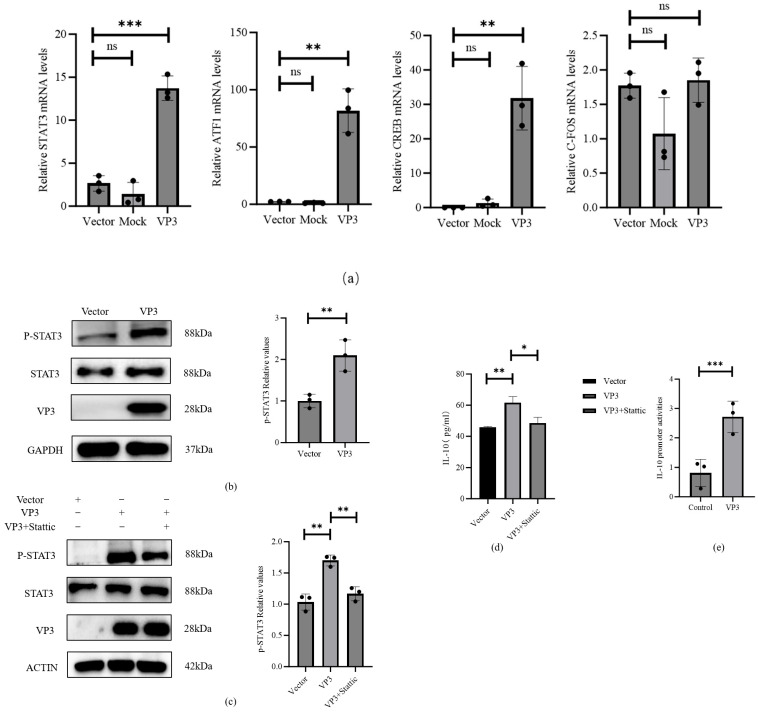
FMDV VP3 modulates STAT3, ATF1, CREB and promotes IL-10 transcription: (**a**) qPCR of STAT3, ATF1, CREB, and c-FOS mRNA; (**b**) Western blot of STAT3 and p-STAT3; (**c**) STAT3 phosphorylation after Stattic treatment; (**d**) IL-10 secretion measured by ELISA after Stattic; (**e**) IL-10 promoter activity assessed by dual-luciferase assay. ns (not significant) *p* > 0.05; * *p* ≤ 0.05; ** *p* ≤ 0.01; *** *p* ≤ 0.001.

**Table 1 viruses-18-00066-t001:** Primers used in this study.

Gene Product	Sense Primer (5′ to 3′)	Antisense Primer (5′ to 3′)
*IL-10*	CGTGGAGGAGGTGAGAGAGTG	TTAGTAGAGTCGTCATCCTGGAAG
*STAT3*	ACAGGATCCTTGACGAGCAC	CTCTCTGAGCCTGTTCCT
*CREB*	CATGGAATCTGGAGCAGACAA	CTGGGCTAATGTGGCAATCT
*C-FOS*	GGCAAGGTGGAACAGTTGTC	CGCTTGGAGTGTGTCAGTCA
*ATF1*	TTGTGCCCAGCAACCAAGTGG	CACGGTCTGTGCAGGGAAAGC

## Data Availability

All data generated for this study are included in the article.

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
