# Peer review of "FMDV VP3 Induces IL-10 Expression in Porcine Macrophages via PI3K Interaction and PI3K/AKT-mTOR Pathway Activation"

_viruses, 2026, doi:10.3390/v18010066_

Round 1
Reviewer 1 Report
Comments and Suggestions for Authors
Yuling Li et al,provide a manuscript of “FMDV VP3 Induces IL-10 Expression in Porcine Macrophages via PI3K Interaction and PI3K/AKT-mTOR Pathway Activation”. The article demonstrated that the foot-and-mouth disease virus (FMDV) structural protein VP3 induces IL-10 secretion in porcine macrophages by activating the PI3K/AKT-mTOR pathway which makes a valuable contribution to understanding FMDV pathogenesis. The manuscript was prepared well although there are still a few of errors in this manuscript. Careful editing should be done before publish.
Minor Issues:
Simplify figure captions (e.g., “Fig. 1. FMDV VP3 induces IL-10 expression in 3D4/21 cells” instead of redundant descriptions).
Non-Standard Figure Numbering: Fig.5-a" uses a hyphen for subfigure. Adopt "Fig.5A" or "Fig.5a
The manuscript alternates between "FMDV-VP3" and "FMDV VP3". Should standardize to "FMDV VP3" (no hyphen) for clarity.
Line10:Remove the redundant "Correspondence:" to "Correspondence: zhangzhidong".
Line22-23:“Mechanistically, VP3 promoted phosphorylation of PI3K, AKT, and mTOR. The specific PI3K inhibitor LY294002 abrogated VP3-induced IL-10 secretion”should be revised to “Mechanistically, VP3 promoted phosphorylation of PI3K, AKT, and mTOR; this effect was abolished by the PI3K inhibitor LY294002, which also abrogated VP3-induced IL-10 secretion”( Combine sentences for coherence).
Line 97: "SensePrimer should be revised to "Sense Primer" in Table 1
Line 105: "Static is a selective STAT3 inhib-itor." , Remove the hyphen: "Static is a selective STAT3 inhibitor."
Line 109-110: The lysed samples were centrifuged at 16,500×g for 10 min at 40C, after which the clarified supernatant was collected.”could be simplified to“The lysed samples were centrifuged at 16,500 × g for 10 min at 40C, and the supernatant was collected.”
Line 152-153: the cells were co-transfected with the pGL-IL-10 and pRL-TK plasmids in complete medium and incubated for 24 hours. Subsequently, the cells were washed again and then transfected with the FMDV-VP3 eukaryotic expression plasmid.”should be changed to “First, cells were co-transfected with pGL-IL-10 and pRL-TK plasmids. After 24 hours, cells were transfected with FMDV-VP3 plasmid.”
Line 228: "Significance versus the vector control group is indicated: **P < 0.05, ***P < 0.01, ****P < 0.0001." should be changed to "Significance versus the vector control group is indicated:*P < 0.05, **P < 0.01, ****P < 0.0001."
Line 278: "high-level interleukin-10 (IL-10) secretion" could be simplified to "high-level IL-10 secretion"
Author Response
Comments 1:Line 109: Please spell out PMSF.
Response 1: Thank you for pointing this out. I agree with this comment. Therefore, I have revised it to PMSF (Line 115).
Comments 2:Line 167: Word 'drives' should be substituted with 'induces'.
Response 2:Thank you for pointing this out. I agree with the comment. Therefore, I have revised it to 'induces' (line 174).
Comments 3:Figure 1 and other relevant figures: Acronym for kilodaltons should be written as 'kDa'. Please increase the font size of Figure 1.
Response 3:Thank you for pointing this out. I agree with the comment. Therefore, I have converted all images to kDa and increased the font size of the images (lines 186-187,203-204,224-225,240-241,264-266).
Comments 4:Lines 183-185: "Dynamic secretion of IL-10 in cell supernatants detected by ELISA. ELISA was used to quantify IL-10 protein levels in supernatants of 3D4/21 cells at 12, 24, and 36 hours post-transfection (hpt)." Please merge these sentences into one coherent statement.
Response 4:Thank you for pointing this out. I agree with the comment. Therefore, I have revised it to: IL-10 secretion in supernatants measured by ELISA at 12,24, and 36 hours post-transfection (lines 189-190).
Comments 5:Lines 186-188: "Transcriptional upregulation of IL-10 mRNA verified by qPCR. qPCR was performed to measure IL-10 mRNA expression levels in FMDV-VP3-transfected versus vector control cells at 12, 24, and 36 hpt." Similarly to above, merge these sentences into one.
Response 5:Thank you for pointing this out. I agree with the comment. Therefore, I have revised it to: IL-10 mRNA levels quantified by qPCR at 12,24, and 36 hours post-transfection.
Comments 1:Simplify figure captions (e.g., “Fig. 1. FMDV VP3 induces IL-10 expression in 3D4/21 cells” instead of redundant descriptions).Non-Standard Figure Numbering: Fig.5-a" uses a hyphen for subfigure. Adopt "Fig.5A" or "Fig.5a
The manuscript alternates between "FMDV-VP3" and "FMDV VP3". Should standardize to "FMDV VP3" (no hyphen) for clarity.
Response 1:Thank you for these valuable suggestions. We have fully addressed the issues as recommended: (1) simplified all figure captions to eliminate redundant descriptions and adopted the concise format as exemplified; (2) unified the subfigure numbering style throughout the manuscript, replacing the hyphenated form (e.g., "Fig.5-a") with "Fig.5a"; (3) standardized the nomenclature of the protein to "FMDV VP3" consistently in the entire text to ensure clarity.
Comments 2:Line10:Remove the redundant "Correspondence:" to "Correspondence: zhangzhidong".
Response 2:We appreciate your careful check. We have deleted the redundant wording in Line 11 and revised the content to "Correspondence: zhangzhidong" as suggested.
Comments 3:Line22-23:“Mechanistically, VP3 promoted phosphorylation of PI3K, AKT, and mTOR. The specific PI3K inhibitor LY294002 abrogated VP3-induced IL-10 secretion”should be revised to “Mechanistically, VP3 promoted phosphorylation of PI3K, AKT, and mTOR; this effect was abolished by the PI3K inhibitor LY294002, which also abrogated VP3-induced IL-10 secretion”( Combine sentences for coherence).
Response 3:Thank you for this constructive advice. We have combined the two sentences in Lines 23–25 according to your suggestion, and the revised text is exactly the version you provided, which has significantly improved the logical coherence of this part.
Comments 4:Line 97: "SensePrimer should be revised to "Sense Primer" in Table 1
Response4:Thanks for pointing out this formatting inconsistency. We have corrected the term in Table 1 (Line 103) from "SensePrimer" to "Sense Primer" to conform to the standard nomenclature format.
Comments 5:Line 105: "Static is a selective STAT3 inhib-itor." , Remove the hyphen: "Static is a selective STAT3 inhibitor."
Response 5:We are grateful for your correction. We have removed the inappropriate hyphen in Line 112 and revised the sentence to "Static is a selective STAT3 inhibitor".
Comments 6:Line 109-110: The lysed samples were centrifuged at 16,500×g for 10 min at 40C, after which the clarified supernatant was collected.”could be simplified to“The lysed samples were centrifuged at 16,500 × g for 10 min at 40C, and the supernatant was collected.”
Response 6:Thank you for the simplification suggestion. We have streamlined the sentence in Lines 115–116 to the concise version you provided, which makes the experimental operation description more succinct and clear.
Comments 7:Line 152-153: the cells were co-transfected with the pGL-IL-10 and pRL-TK plasmids in complete medium and incubated for 24 hours. Subsequently, the cells were washed again and then transfected with the FMDV-VP3 eukaryotic expression plasmid.”should be changed to “First, cells were co-transfected with pGL-IL-10 and pRL-TK plasmids. After 24 hours, cells were transfected with FMDV-VP3 plasmid.”
Response 7:We agree with this concise revision proposal. We have revised the content in Lines 160–161 to the exact version you recommended, which effectively condenses the description of the transfection procedure while retaining all key information.
Comments 8:Line 228: "Significance versus the vector control group is indicated: **P < 0.05, ***P < 0.01, ****P < 0.0001." should be changed to "Significance versus the vector control group is indicated:*P < 0.05, **P < 0.01, ****P < 0.0001."
Line 278: "high-level interleukin-10 (IL-10) secretion" could be simplified to "high-level IL-10 secretion"
Response 8:Thank you for these two detailed revision comments. We have made the corresponding corrections as follows: (1) Adjusted the significance annotation symbols in Line 228 to the standardized format you provided; (2) Simplified the expression in Line 278 from "high-level interleukin-10 (IL-10) secretion" to "high-level IL-10 secretion" to eliminate redundant terminology.
Correction and Optimization of Western Blot Images
We sincerely apologize for the accidental misuse of internal reference images in the initial submission. To improve the accuracy and clarity of data presentation, we have made two types of revisions to the Western Blot (WB) figures, with all core conclusions remaining unchanged. The complete original data associated with these experiments has been submitted to the editors for verification.
Correction of internal reference bandsThe internal reference bands in Figure 1 were incorrectly matched in the initial version. We have replaced them with the correct internal reference bands derived from the same batch of experiments, ensuring the accuracy of protein expression normalization (Lines 185–187).
Optimization of representative WB imagesFor Figures 2 and 3, we substituted the original bands with clearer and more representative results from biological replicates of the identical experiment. Specifically, Figure 2 shows the expression of p-PI3K and AKT, while Figure 3 displays p-PI3K levels. Notably, the conclusions drawn from the revised data are fully consistent with those in the original manuscript and do not alter any core experimental arguments (Lines 203–204, 224–225).
All revised WB images were generated under the same experimental conditions and biological repeats as described in the Materials and Methods section. The raw WB membrane images and full-length uncropped blot scans have been uploaded as Supplementary Materials for peer review. We confirm that no data has been fabricated or manipulated, and all modifications strictly adhere to the journal’s guidelines for data presentation. We appreciate the opportunity to refine our manuscript and thank the reviewers/editors for their careful oversight.

Reviewer 2 Report
Comments and Suggestions for Authors
This manuscript reports new information on the Foot and mouth disease virus (FMDV)-VP3 protein-induced expression of IL-10 in porcine macrophages via PI3K interaction. The study is well-designed, and the methodology is clearly described. Results are presented appropriately along with data analyses. The study conclusions support the study hypothesis. This reviewer has a few minor suggestions for the authors, as follows:
Line 109: Please spell out PMSF.
Line 167: Word 'drives' should be substituted with 'induces'.
Figure 1 and other relevant figures: Acronym for kilodaltons should be written as 'kDa'. Please increase the font size of Figure 1.
Lines 183-185: "Dynamic secretion of IL-10 in cell supernatants detected by ELISA. ELISA was used to quantify IL-10 protein levels in supernatants of 3D4/21 cells at 12, 24, and 36 hours post-transfection (hpt)." Please merge these sentences into one coherent statement.
Lines 186-188: "Transcriptional upregulation of IL-10 mRNA verified by qPCR. qPCR was performed to measure IL-10 mRNA expression levels in FMDV-VP3-transfected versus vector control cells at 12, 24, and 36 hpt." Similarly to above, merge these sentences into one.
Author Response
Comments 1:Line 109: Please spell out PMSF.
Response 1: Thank you for pointing this out. I agree with this comment. Therefore, I have revised it to PMSF (Line 115).
Comments 2:Line 167: Word 'drives' should be substituted with 'induces'.
Response 2:Thank you for pointing this out. I agree with the comment. Therefore, I have revised it to 'induces' (line 174).
Comments 3:Figure 1 and other relevant figures: Acronym for kilodaltons should be written as 'kDa'. Please increase the font size of Figure 1.
Response 3:Thank you for pointing this out. I agree with the comment. Therefore, I have converted all images to kDa and increased the font size of the images (lines 186-187,203-204,224-225,240-241,264-266).
Comments 4:Lines 183-185: "Dynamic secretion of IL-10 in cell supernatants detected by ELISA. ELISA was used to quantify IL-10 protein levels in supernatants of 3D4/21 cells at 12, 24, and 36 hours post-transfection (hpt)." Please merge these sentences into one coherent statement.
Response 4:Thank you for pointing this out. I agree with the comment. Therefore, I have revised it to: IL-10 secretion in supernatants measured by ELISA at 12,24, and 36 hours post-transfection (lines 189-190).
Comments 5:Lines 186-188: "Transcriptional upregulation of IL-10 mRNA verified by qPCR. qPCR was performed to measure IL-10 mRNA expression levels in FMDV-VP3-transfected versus vector control cells at 12, 24, and 36 hpt." Similarly to above, merge these sentences into one.
Response 5:Thank you for pointing this out. I agree with the comment. Therefore, I have revised it to: IL-10 mRNA levels quantified by qPCR at 12,24, and 36 hours post-transfection.
Correction and Optimization of Western Blot Images
We sincerely apologize for the accidental misuse of internal reference images in the initial submission. To improve the accuracy and clarity of data presentation, we have made two types of revisions to the Western Blot (WB) figures, with all core conclusions remaining unchanged. The complete original data associated with these experiments has been submitted to the editors for verification.
Correction of internal reference bandsThe internal reference bands in Figure 1 were incorrectly matched in the initial version. We have replaced them with the correct internal reference bands derived from the same batch of experiments, ensuring the accuracy of protein expression normalization (Lines 185–187).
Optimization of representative WB imagesFor Figures 2 and 3, we substituted the original bands with clearer and more representative results from biological replicates of the identical experiment. Specifically, Figure 2 shows the expression of p-PI3K and AKT, while Figure 3 displays p-PI3K levels. Notably, the conclusions drawn from the revised data are fully consistent with those in the original manuscript and do not alter any core experimental arguments (Lines 203–204, 224–225).
All revised WB images were generated under the same experimental conditions and biological repeats as described in the Materials and Methods section. The raw WB membrane images and full-length uncropped blot scans have been uploaded as Supplementary Materials for peer review. We confirm that no data has been fabricated or manipulated, and all modifications strictly adhere to the journal’s guidelines for data presentation. We appreciate the opportunity to refine our manuscript and thank the reviewers/editors for their careful oversight.
